# ISCUTE: INSTANCE SEGMENTATION OF CABLES USING TEXT EMBEDDING

## ABSTRACT

In the field of robotics and automation, conventional object recognition and instance segmentation methods face a formidable challenge when it comes to perceiving Deformable Linear Objects (DLOs) like wires, cables, and flexible tubes. This challenge arises primarily from the lack of distinct attributes such as shape, color, and texture, which calls for tailored solutions to achieve precise identification. In this work, we propose a foundation model-based DLO instance segmentation technique that is text-promptable and user-friendly. Specifically, our approach combines the text-conditioned semantic segmentation capabilities of CLIPSeg model with the zero-shot generalization capabilities of Segment Anything Model (SAM). We show that our method exceeds SOTA performance on DLO instance segmentation, achieving a mIoU of $91.21\%$. We also introduce a rich and diverse DLO-specific dataset for instance segmentation.

## 1 INTRODUCTION

Deformable Linear Objects (DLOs), encompassing cables, wires, ropes, and elastic tubes, are commonly found in domestic and industrial settings (Keipour et al., 2022b; Sanchez et al., 2018). Despite their widespread presence in these environments, DLOs present significant challenges to automated robotic systems, especially in perception and manipulation, as discussed by Cop et al. (2021). In terms of perception, the difficulty arises from the absence of distinct shapes, colors, textures, and prominent features, which are essential factors for precise object perception and recognition.

Over the past few years, there has been a notable emergence of approaches customized for DLOs. State-of-the-art (SOTA) instance segmentation methods such as RT-DLO (Caporali et al., 2023) and mBEST (Choi et al., 2023) use novel and remarkable approaches influenced by classic concepts from graph topology or by bending energy to segment these challenging objects accurately. Nevertheless, none of these methods excel in handling real and complex scenarios, nor do they incorporate prompt-based control functionality for segmentation, such as text prompts, which could enhance user accessibility.

In the domain of purely computer vision-based approaches, the Segment Anything Model (SAM; Kirillov et al., 2023) is one of the most notable segmentation models in recent years. As a foundation model, it showcases remarkable generalization capabilities across various downstream segmentation tasks, using smart prompt engineering (Bomasani & Others, 2021). However, SAM's utility is limited to manual, counterintuitive prompts in the form of points, masks, or bounding boxes, with basic, proof-of-concept text prompting. On the other hand, in the domain of deep vision-language fusion, CLIPSeg (Lüddecke & Ecker, 2022) presents a text-promptable semantic segmentation model. However, this model does not extend its capabilities to instance segmentation.

In this paper, we present a novel adapter model that facilitates as a communication channel between the CLIPSeg model and SAM, as illustrated in Figure 1. Specifically, our model is capable of transforming semantic mask information corresponding to a text prompt to batches of point prompt embedding vectors for SAM, which processes them to generate instance segmentation masks.

In our method, we exhibit the following novelties:

1. A novel prompt encoding network that converts semantic information obtained from text prompts to point prompts' encoding that SAM can decode.

2. A classifier network to filter out duplicate or low-quality masks generated by SAM.

3. A CAD-generated, fully annotated, DLO-specific dataset consisting of about 30k high-resolution images of cables in industrial settings.

Our method also improves the SOTA in DLO segmentation, achieving $mIoU = 91.21\%$ on the generated dataset, compared to RT-DLO ($mIoU = 50.13\%$). Furthermore, we show our model's strong zero-shot generalization capabilities to datasets presented in RT-DLO (Caporali et al., 2023) and mBEST (Choi et al., 2023). Finally, our method offers a user-friendly approach to the problem, requiring only an image and a text prompt to obtain one-shot instance segmentation masks of cables.

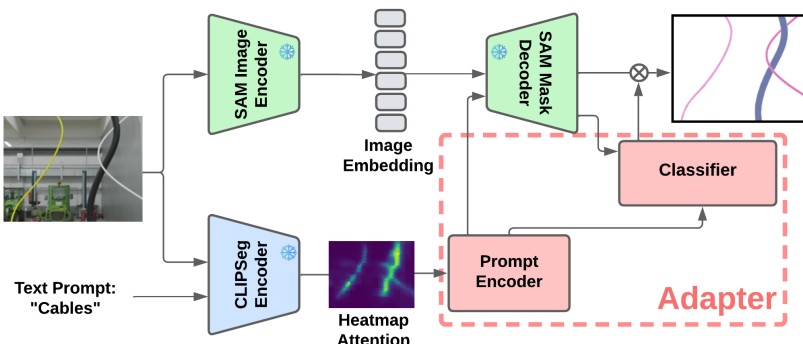

Figure 1: Overview of the full pipeline - blocks in red represent our additions

## 2 RELATED WORK

In this section, we review related works of DLOs instance segmentation and methods we use in our approach.

### 2.1 DLOS INSTANCE SEGMENTATION

DLO detection algorithms have been introduced with distinctive methodologies that are necessary to overcome the challenges associated with the problem. Zanella et al. (2021) first presented a Convolutional Neural Network (CNN; LeCun et al., 1995) based approach to perform semantic segmentation of DLOs. Yan et al. (2019) and Keipour et al. (2022a) pioneered data-driven techniques for DLO instance segmentation, using neural networks to reconstruct DLO topology and fitting curvatures and distances for a continuous DLO, respectively. However, these methods were limited to a single DLO.

*Ariadne* (De Gregorio et al., 2018) and it is immediate successor *Ariadne+* (Caporali et al., 2022b), segment DLOs into superpixels (Achanta et al.), followed by path traversal to generate instance masks. *Ariadne+* provided a DeepLabV3+ (Chen et al., 2018) based model to extract a semantic mask before superpixel processing. This allowed the authors to forego *Ariadne's* limiting assumption about having a uniformly colored DLO and requiring the manual selection of endpoints. However, both Ariadne and Ariadne+ encounter a significant challenge due to their reliance on numerous hyperparameters .

FASTDLO (Caporali et al., 2022a) and RT-DLO (Caporali et al., 2023), approach DLOs instance segmentation using skeletonization of semantic masks, instead of superpixel generation. This allows both algorithms to operate much faster because the slow process of superpixel generation is non-essential. In FASTDLO, the skeletonized semantic mask is processed using path traversal, and a neural network is used to process intersections. Contrastively, RT-DLO employs a graph node sampling algorithm from DLO centerlines using the semantic mask. This is followed by topological reasoning-based edge selection and path determination. Although both algorithms offer either near-real-time or real-time performance (depending on the number of DLOs and intersections in the image), they are sensitive to noise introduced by centerline sampling and scene-dependent hyperparameter tuning.

Another recent work, mBEST (Choi et al., 2023), involves fitting a spline to minimize bending energy while traversing skeleton pixels. This method has shown impressive results compared to all the aforementioned works, exhibiting a unique solution to the problem, but it is limited to fewer than two intersections between different DLOs and relies on a manually tuned threshold that depends on the number of DLOs in the scene.

All the aforementioned approaches, including the current SOTA baselines, RT-DLO (Caporali et al., 2023) and mBEST (Choi et al., 2023) exhibit sensitivity to scenarios such as occlusions, cables intersecting near image edges, and small cables positioned at the edges of the image, which are commonly encountered scenes in industrial settings. Furthermore, these methods face challenges when transferring to other datasets and are limited to images with specific operating conditions. Our approach is designed to address these challenges through the utilization of a diverse and comprehensive generated dataset as well as the generalization power of foundation models. Furthermore, to the best of our knowledge, this is the first approach that combines DLO instance segmentation with text prompts, offering a user-friendly solution.

## 2.2 Foundation Models

Introduced by Kirillov et al. (2023), SAM is a powerful foundation model that shows remarkable performance on a myriad of downstream tasks. It shows promising performance on datasets containing DLOs that have historically been difficult to segment. The generalization power of this model lies in its prompting capabilities, specifically using points, boxes, or masks to direct the segmentation. SAM outperforms other promptable segmentation models such as FocalClick (Chen et al.), PhraseClick (Ding et al.), PseudoClick (Liu et al., 2022b) and SimpleClick (Liu et al., 2022a) in the generalizability and quality of segmentation. The authors of SAM also present a proof-of-concept for text-based prompting, but this method still requires manual point prompt assistance to give the same high-quality segmentation masks. Box prompts are not conducive to one-shot instance segmentation, as a single bounding box can encompass multiple DLOs, as illustrated in Fig. 2c. Single point-prompting displays a strong potential for performance (Fig. 2b). However, it struggles to perform effectively in complex scenarios (Fig. 2a). Manually inputting multiple points per cable proves inefficient for process automation in an industrial context because it necessitates human intervention. In our work, we harness the capabilities of the SAM model for segmentation and its ability to use point prompts. We replaced its prompt encoder, capable of converting text prompts into point prompts' embedding. Moreover, our model performs instance segmentation in one forward pass without the need for sequential mask refinement as in Kirillov et al. (2023). Additionally, we also introduce a network to filter out duplicate and low-quality masks.

Multi-modal data processing such as text-image, text-video, and image-audio has gained impetus in recent years following the advent of mode-fusion models such as CLIP (Radford et al., 2021) and GLIP (Li et al., 2021). GroundingDINO (Liu et al., 2023) uses BERT (Devlin et al.) and either ViT (Dosovitskiy et al., 2020) or Swin-Transformer (Liu et al., 2021) to perform text conditioned object detection. The approach is a SOTA object detector but relies on bounding boxes, proving less effective for DLOs (Fig. 2c).

A notable counter example is CLIPSeg (Lüddecke & Ecker, 2021), that leverages a CLIP (Radford et al., 2021) backbone followed by a FiLM layer (Perez et al., 2017) and finally a UNet (Ronneberger et al., 2015) inspired decoder to generate semantic segmentation based on text prompts (Fig. 1). Important to our application, the penultimate layer's embedding space in CLIPSeg represents a spatially oriented attention heatmap corresponding to a given text prompt.

Another recent trend in computer-vision research is *adapters* that have been proposed for incremental learning and domain adaptation (Chen et al., 2022; Rebuffi et al., 2017; Selvaraj et al., 2023). Some adapters like VL-Adapter (Sung et al., 2021), CLIP-Adapter (Gao et al., 2021), Tip-Adapter (Zhang et al., 2021) use CLIP to transfer pre-trained knowledge to downstream tasks. The method we propose in this work is also inspired by some ideas from research in adapters.

## 3 Methods

In this section, we describe the core method behind our approach.

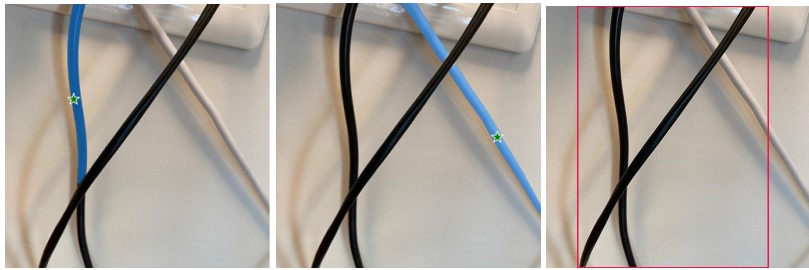

(a) An example of unsuc-cessful segmentation with a single point.

(b) An example of success-ful segmentation with a sin-gle point

(c) An example of box prompt for DLO segmenta-tion.

Figure 2: Issues with using SAM out-of-the-box

## 3.1 THE ISCUTE MODEL

The adapter model we propose consists of 2 main networks (as depicted in Fig. 1):

1. **Prompt encoder network** - This network extracts batches of point prompts from CLIPSeg's embedding space. It can be controlled using 2 hyperparameters: $N$ (the number of prompt batches) and $N_p$ (the number of points in each batch).

2. **Classifier network** - A binary classifier network that labels whether a particular mask from the generated $N$ masks should appear in the final instance segmentation.

In the following, we detail each of these networks. We perform ablation studies to justify our choice of architecture. The results of these ablation studies can be found in Appendix C. We note that we neither fine-tune CLIPSeg nor fine-tune SAM, and the only trained component is the adapter. The motivation for this is twofold. First, we would like to refrain from degrading the original performance of both CLIPSeg and SAM. Second, it is more efficient in many cases to train only an adapter instead of foundation models.

### 3.1.1 PROMPT ENCODER NETWORK

The Prompt Encoder Network's goal is to efficiently convert the embedding space generated by CLIPSeg into batches of point prompt embedding that SAM's mask decoder can accurately deci-pher, as depicted in Figure 3. This process of selecting the appropriate point prompts relies on the following stages:

1. Identification of critical patches.
2. Determining an optimal threshold based on patch importance and filtering out the rest.
3. Categorizing each selected point as foreground, background, or no-point to leverage SAM's capabilities.

CLIPSeg generates a $22 \times 22 \times 64$ embedding tensor, which embeds a semantic mask that aligns with the input image spatially and is conditioned on text. To maintain a consistent embedding size throughout the pipeline, we employ an MLP (bottom left MLP in Fig. 3) to upscale the 64-dimensional embedding to 256 dimensions, followed by a self-attention layer, which learns inter-patch correlations to focus on the relevant patches. CLIPSeg's embedding output is enhanced with Dense Positional Encoding (DPE) to ensure that the self-attention layer has access to crucial ge-ometric information. To this end, the DPE values are added to the embedding vector even after participating in the self-attention layer. To generate our DPE, we use an identical frequency matrix as SAM. This ensures that every element within each vector of the DPE conveys consistent informa-tion, that is aligned with what SAM's decoder has been trained to interpret. This output is processed by an MLP that acts like a filter and aims to retain only the most important patches. After the im-portant patches are extracted, we pass them into the Sampler Attention (pink layer in Fig. 3). This process resembles standard scoring, where the interaction of the keys and queries determines which positional encoding vector to select from the DPE. Here, the self-attended patches act as the keys,

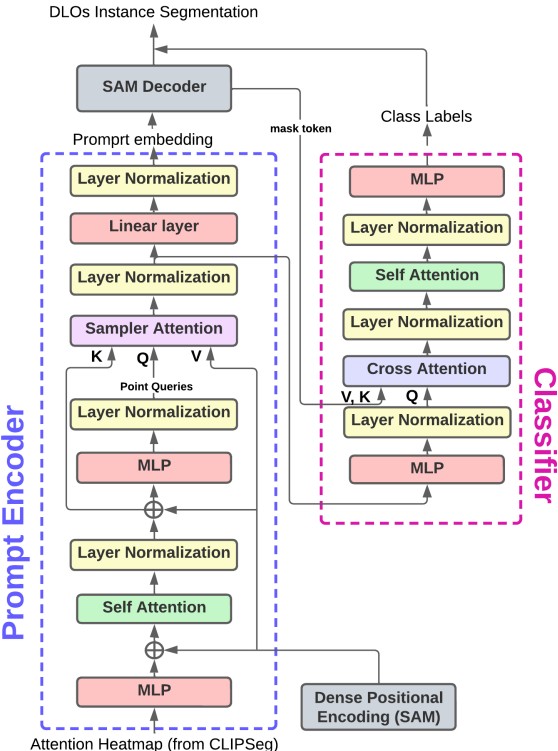

Figure 3: The ISCUTE adapter: on the left, the architecture of the prompt encoder network is outlined (indicated by the purple dashed line), while on the right, the classifier network architecture is detailed (represented by the pink dashed line).

and the DPE tensor as the values. The output of this layer is the positional encoding of the points we wish to prompt SAM with. Finally, a single Linear layer is applied to this output, to categorize the chosen point as foreground/background/no-point. This layer aims to mimic the labeling protocol from SAM, which employs a standard affine transformation, where a learned embedding is added to each point embedding corresponding to its category.

It is important to highlight that the queries are reintroduced to the updated tokens whenever they engage in an attention layer. This allows for a strong dependence on both the queries that carry the semantic information and the values that carry the geometric information. Moreover, layer normalization (Ba et al., 2016) is applied after each attention layer, as recommended by Vaswani et al. (2023). Furthermore, our DPE, like SAM, takes inspiration from Fourier features (Tancik et al., 2020) alongside classical techniques from digital communication (Ling, 2017), utilizing the same frequency matrix as SAM instead of a randomly generated one.

### 3.1.2 MASK CLASSIFICATION NETWORK

In previous works, such as DETR (Carion et al., 2020) and MaskFormer (Cheng et al., 2021), the authors train a classifier network along with the box regression model to classify which object is contained within the box. Both of these works introduced an additional *no-object class*, to filter out duplicate or erroneous boxes/masks during prediction. Inspired by them, we developed a binary classifier network for this purpose.

This binary classifier model is composed of an MLP, one cross-attention block, followed by one self-attention block, and an MLP, as illustrated in Figure 3. First, the sampled point embeddings (from our prompt encoder) are transformed using an MLP. Second, a cross-attention block operates on these transformed embedding generated by our model, which encode text-conditioned information, and on the mask tokens produced by SAM, which encapsulate submask-related details. This

interaction results in an embedding that fuses both types of information for each generated submask. Subsequently, the queries are combined with the output to reintroduce textual information to these *classifier tokens*. These classifier tokens then undergo a self-attention layer followed by the MLP to yield binary classifications.

## 3.2 TRAINING PROTOCOL

The full architectural framework of our model, illustrated in Figure 1, accepts a single RGB image and a text prompt as input, delivering an instance segmentation mask for each DLO in the image. Inspired by the work of Cheng et al. (2021), we employ a bipartite matching algorithm (Kuhn, 1955) to establish the optimal correspondence between the generated masks and ground-truth submasks to train the prompt encoder. We use a combination of the focal loss (Lin et al., 2017) and the DICE loss (Milletari et al., 2016) in the ratio of $20 : 1$, as recommended by Kirillov et al. (2023).

The binary classification labels are extracted from the output of the bipartite matching algorithm. If a mask successfully matches, then it is labeled as $1$; else, it is labeled as $0$. We use the binary cross-entropy loss to train this network. To balance the distribution of 0s and 1s in the dataset, we introduce a weight for positive labels that we set to $3$.

The most time-intensive step within the SAM workflow involves generating the image embedding. However, once this image embedding is created, it can be used repeatedly to produce multiple submasks as needed. This approach significantly speeds up the process and can be executed in a single step. To enhance training efficiency using this method, we set the training batch size to just $1$. This configuration enables us to form batches of prompts, with each DLO in the scene associated with an individual prompt. We place a cap of $N = 11$ prompts for each image and limit the number of points within each prompt to $N_p = 3$. Specific details about the training process can be found in Appendix A.

## 4 EXPERIMENTS

### 4.1 THE CABLES DATASET

The dataset used for training, testing, and validation was generated using Blender[1], a 3D rendering software. It comprises images depicting various cables in diverse thicknesses, shapes, sizes, and colors within an industrial context, exemplified in Figures 1 and 4. The dataset consists of 22k training images, along with 3k images, each designated for validation and testing. Each image has a resolution of 1920x1080, and each DLO within an image is accompanied by a separate mask, referred to as a submask in this work. This pioneering DLO-specific dataset encompasses an extensive range of unique scenarios and cable variations. We anticipate that granting access to this comprehensive dataset will significantly advance numerous applications in the field of DLO perception within industrial environments. For more details and sample images from the dataset, refer to the Appendix B. This dataset will be made publicly available on acceptance.

### 4.2 BASELINE EXPERIMENTS

In the process of designing the model and configuring the best hyperparameters, we carried out multiple experiments. The initial experiment involved selecting the number of points per prompt, denoted as $N_p$. As previously mentioned, the point prompt within the SAM framework can encompass more than just a single point, with each point being labeled as either foreground or background. In the SAM paper (Kirillov et al., 2023), the authors explored various values for the number of points and documented the resulting increase in Intersection over Union (IoU) for the generated masks. They observed diminishing returns after reaching 3 points. However, it was necessary to reevaluate this parameter for one-shot applications. We conducted this using the same base model but with different values for $N_p$, specifically $N_p = 2, 3, 4$. Our findings led us to the conclusion that $N_p = 3$ is indeed the optimal number of points per prompt, and we adopted this value for all subsequent experiments. The second experiment focuses on SAM's capacity to generate multiple masks, specifically, three masks, per prompt. We examined and compared this multi-mask output

---

[1]https://www.blender.org/

with the single-mask output in terms of the mean IoU (mIoU). Our findings indicated that employing multi-mask output led to only a minor improvement at the cost of significantly higher memory utilization. Consequently, for all our subsequent experiments involving SAM, we opted to disable the multi-mask functionality.

### 4.3 QUANTITATIVE EXPERIMENTS

In these experiments, our primary objective is to compare our model's DLO instance segmentation results with those of SOTA baselines while also assessing the model's limitations using the *Oracle* method, inspired by Kirillov et al. (2023), which involves removing the classifier network during testing and employing bipartite matching to filter out duplicate and low-quality masks. This method allows us to independently evaluate our prompt encoder and classifier networks, providing insights into the performance of foundation models under ideal conditions, with the important note that our model has access to ground-truth annotations during Oracle tests.

We additionally test the following characteristics - the overall quality of the generated instance masks, zero-shot generalization capabilities, and the effect of image augmentation on the generalization. In Table 1, we explore the effects of image augmentation and the oracle method. Adhering to recent testing protocols in the domain of DLO instance segmentation, as conducted by Choi et al. (2023) in mBEST, we use the DICE score as a metric to analyze the performance of our model. The comparative results are portrayed in Table 2. These tests exhibit the zero-shot transfer capabilities of our models to the testing datasets discussed in mBEST (Choi et al., 2023).

## 5 RESULTS

### 5.1 QUANTITATIVE RESULTS

We summarize the comparison of results of different configurations of our model on our test dataset in Table 1. As a reference, we tested the RT-DLO algorithm (Caporali et al., 2023) on our test dataset. It achieved an $mIoU = 50.13\%$. The Oracle tests show upper bound on the performance our method can achieve. This upper bound is related to the limitations of CLIPSeg to produce good semantic masks given a text prompt and the limitations of SAM to segment a particular object in the scene given a set of points as a prompt.

| Model Configuration | Test data performance | | Augmentation | Oracle |
|---|---|---|---|---|
| | mIoU [%] | DICE [%] | | |
| **A** (Aug only) | 90.64 | 99.78 | Y | N |
| **A+O** (Aug+Oracle) | 92.10 | 99.80 | Y | Y |
| **B+O** (Base+Oracle) | **92.51** | **99.82** | N | Y |
| **B** (Base) | 91.21 | 99.77 | N | N |

Table 1: Comparison of our model configurations (values in bold signify best performance).

Table 2 shows the DICE scores of our model compared to current SOTA baselines: Ariadne+ (Caporali et al., 2022b), FASTDLO (Caporali et al., 2022a), RT-DLO (Caporali et al., 2023) and mBEST (Choi et al., 2023). The datasets C1, C2, C3 are published by Caporali et al. (2023), while the datasets S1, S2, S3 are published by Choi et al. (2023).

In the *no Oracle* setting, we observe that the model generalizes better if it is trained using augmentations in the dataset, as expected. ISCUTE exhibits strong zero-shot generalization to all the datasets. Specifically for C1, C2, and C3, our model exceeds SOTA performance even in the base configuration. Furthermore, in the case of Oracle configuration, we see that our proposed method outperforms all SOTA baselines on all datasets, showing that the major bottleneck in our full model is the mask classifier network.

To assess generalizability across various text prompts, we tested our model on prompts like "wires" and "cords," in addition to our training prompt, "cables," which specifically refers to DLOs. Table 3 showcases the performance of these prompts when assessed on the same model using Oracle.

| | DICE[%] | | | | | | | |
|---|---|---|---|---|---|---|---|---|
| | | | | | **ISCUTE (Ours)** | | | |
| Dataset | Ariadne+ | FASTDLO | RT-DLO | mBEST | A | A+O | B+O | B |
| C1 | 88.30 | 89.82 | 90.31 | 91.08 | 97.03 | 98.85 | **98.86** | 94.55 |
| C2 | 91.03 | 91.45 | 91.10 | 92.17 | 96.91 | 98.69 | **98.70** | 96.12 |
| C3 | 86.13 | 86.55 | 87.27 | 89.69 | 97.13 | 98.81 | **98.90** | 90.26 |
| S1 | 97.24 | 87.91 | 96.72 | 98.21 | 97.36 | 98.43 | **98.60** | 93.08 |
| S2 | 96.81 | 88.92 | 94.91 | 97.10 | 97.71 | **98.54** | 98.42 | 97.01 |
| S3 | 96.28 | 90.24 | 94.12 | 96.98 | 96.69 | **98.79** | 98.58 | 95.83 |

Table 2: DICE score result from comparison for out-of-dataset, zero-shot transfer. Values in bold signify the best performance

Remarkably, our model demonstrates the ability to effectively generalize across different prompts zero-shot, producing results comparable to the base prompt, "cables."

| Text Prompt | mIoU [%] | DICE [%] |
|---|---|---|
| cables (baseline) | 92.51 | 99.82 |
| wires | 89.6 | 99.77 |
| cords | 89.3 | 99.64 |

Table 3: Generalizability Across Text Prompts: Demonstrates results comparable to the base prompt "cables" through zero-shot transfer.

## 5.2 QUALITATIVE RESULTS

In this section, we will present the actual instance masks generated by our model. We used the configuration that was trained on augmentations (A) to generate all these results (without Oracle).

Figure 4 presents the instance masks generated by ISCUTE in various complex scenarios, along with the results of RT-DLO on the same image set for comparison. These scenarios encompass occlusions, DLOs with identical colors, varying thicknesses, small DLOs in the corners, a high density of cables in a single image, and real-world scenarios. Our model shows impressive performance across all of these scenarios.

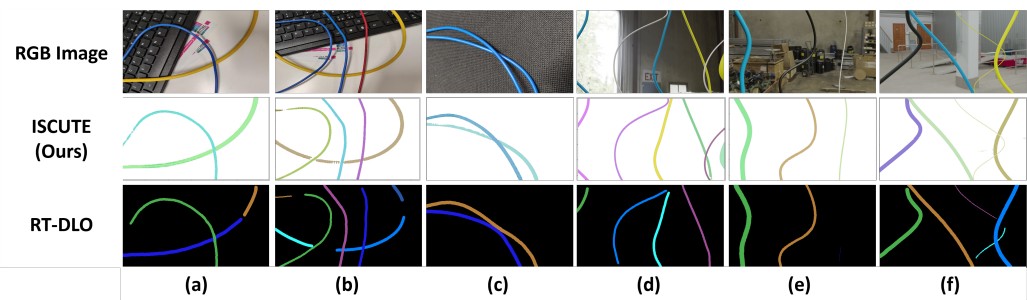

Figure 4: Qualitative comparison in specific scenarios. Each scenario demonstrates the following: (a) and (b) real images, (c) identical colors, (d) a high density of cables in a single image, and (e) and (f) small DLOs at the edge of the image with varying thicknesses.

In Figure 5, we show the instance masks generated by ISCUTE compared to the current SOTA baselines on an external dataset. It is evident that our model demonstrates remarkable zero-shot transfer capabilities when applied to data distributions different from its training data.

Finally, we test our model on real images taken around our lab using a smartphone camera. The final output masks are shown in Figure 6. We see that our model generalizes to real-world, complex

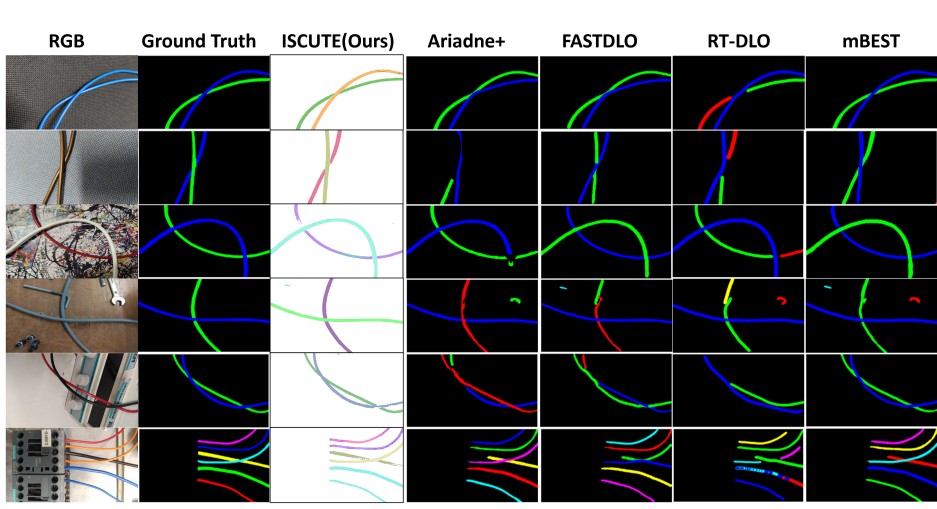

Figure 5: A qualitative comparison of our model vs. the SOTA baselines

scenarios zero-shot. However, because of an imperfect classifier, it misses some submasks in certain complex situations. We report more generated instance masks in Appendix D.

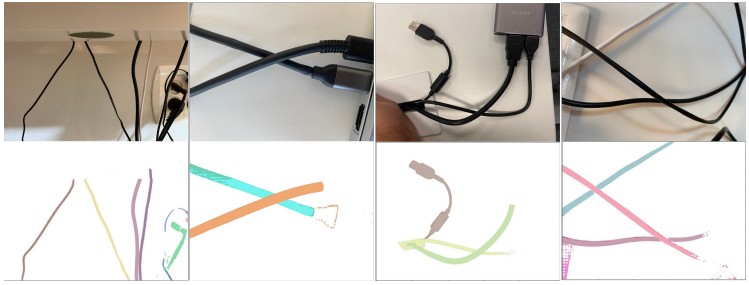

Figure 6: Qualitative results on real-world, complex scenarios zero-shot

# 6 CONCLUSIONS AND FUTURE WORK

We introduced a novel method for DLO instance segmentation, leveraging SAM, a robust foundation model for segmentation, alongside CLIPSeg, which combines text prompts with semantic segmentation. We presented and tested an adapter model that converts text-conditioned semantic masks embedding into point prompts for SAM and introduced a novel approach to automatically filter duplicate and low-quality instance masks generated by SAM, both conditioned on text embedding, potentially enabling adaptation to standard rigid-object instance segmentation with text prompts. This work is an avenue for future research.

Our results show that our model performs exceptionally on our test dataset and exhibits exceptional zero-shot transfer capability to other DLO-specific datasets. However, the final outcome of the segmentation is upper-bounded by the performance of SAM and CLIPSeg, as we observe from the Oracle tests (Refer to Appendix E for a detailed discussion). The performance with Oracle configuration exhibits the strength of our prompting network, given an optimal classifier network. In the current state (without Oracle), our classifier network limits the highest performance our model can achieve without access to the ground truth. Improving our classifier network is an ongoing line of work.

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

## A   TRAINING DETAILS

In our method, we chose the ViT-L/16-based model of SAM (Kirillov et al., 2023) to attempt to balance the speed-accuracy trade-off. We observed that this model gave high-quality segmentation masks while being $\sim 2\times$ smaller compared to ViT-H/16 in terms of the number of parameters. On the other hand, for the CLIPSeg (Lüddecke & Ecker, 2021) model, we used the ViT-B/16 based model, with $reduce\_dim = 64$. Throughout our training process, we freeze the weights and biases of both foundational models, SAM and CLIPSeg.

Throughout the training phase, our text prompt remains "cables," with the aim of obtaining instance segmentation for *all* the cables within the image. During training, we conduct augmentations applied to the input images. These augmentations encompass random grayscale conversion, color jitter, and patch-based and global Gaussian blurring and noise. In terms of computing, our model is trained using 2 NVIDIA A5000 GPUs, each equipped with 32 GB of memory. All testing is carried out on a single NVIDIA RTX 2080 GPU with 16 GB of memory.

Our training procedure employs a learning rate warm-up spanning 5 epochs, followed by a cosine decay. The peak learning rate is set to $lr = 0.0008$, in line with recommendations from Kirillov et al. (2023). We employ the default *AdamW* optimizer from Paszke et al. (2019) with a weight decay of 0.01. The convergence graphs and learning rate profile of all the models can be seen in Figure 7. Figure 8 displays the binary classification accuracy and mIoU computed on the validation dataset during training. No smoothing was applied in creating the plots.

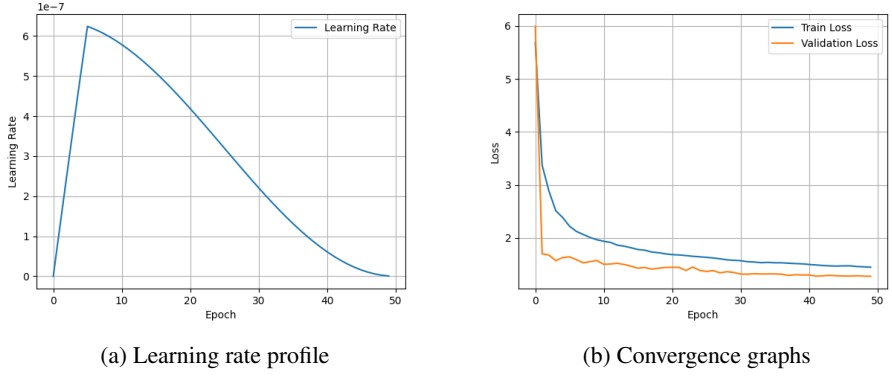

(a) Learning rate profile         (b) Convergence graphs

Figure 7: Learning rate profile and convergence graphs

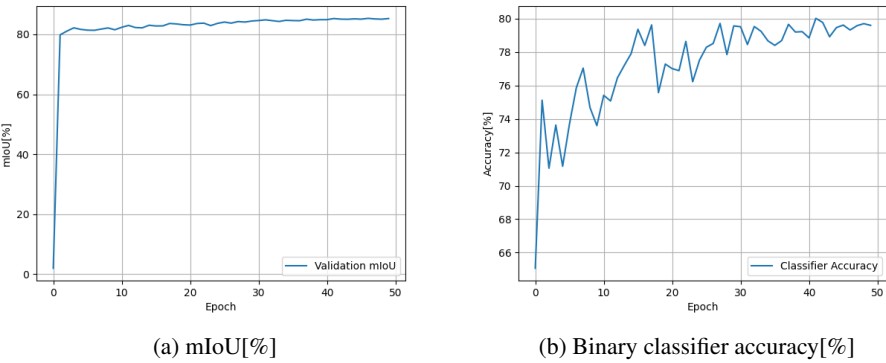

(a) mIoU[%]         (b) Binary classifier accuracy[%]

Figure 8: Binary classification accuracy and mIoU

A brief summary of all the *hyperparameters* can be seen the Table 4

| Hyperparameter | Value |
|---|---|
| Number of epochs | 50 |
| Max learning rate | $8 \times 10^{-4}$ |
| Learning rate warmup | 5 (epochs) |
| Optimizer | *AdamW* |
| Optimizer weight decay | 0.01 |
| Batch size | 1 |
| Attention dropout | 0.5 |
| Number of prompts per batch ($N$) | 11 |
| Number of points per prompt ($N_p$) | 3 |
| Number of attention heads (for all models) | 8 |
| Embedding dimension | 256 |
| SAM model type | *ViT-L/16* (frozen) |
| CLIPSeg model type | *ViT-B/16* (frozen) |
| Focal loss weight | 20 |
| DICE loss weight | 1 |
| Positive label weight (binary cross-entropy) | 3 |
| Classifier MLP activation | *ReLU* |
| Prompt encoder MLP activation | *GELU* |
| Train dataset size | 20038 |
| Validation dataset size | 3220 |
| Test dataset size | 3233 |
| Image size | $(1920 \times 1080)$ |
| Total number of parameters (including foundation models) | 466M |
| Trainable parameters | 3.3M |

Table 4: Hyperparameters

## B GENERATED DATASET

The dataset we presented contains 20038 train images, 3220 validation images, and 3233 test images. Each image we provide is accompanied by its corresponding semantic mask and binary submasks for each DLO in the image. The images are located in {train,test,val}/RGB, and named as train/RGB/00000_0001.png, train/RGB/00001_0001.png, and so on. In the {train, test, val}/Masks folder, we have a sub-folder containing the binary submasks for each corresponding RGB image. For example, train/Masks/00000 contains all the submasks corresponding to train/RGB/00000_0001.png. Additionally, the semantic mask for train/RGB/00000_0001.png is called train/Masks/00000_mask.png. The folder structure can be seen in Figure 9.

Each image and its corresponding masks are $1920 \times 1080$ in resolution. The number of cables, their thickness, color, and bending profile are randomly generated for each image. There are 4 possible colors for the cables - *cyan, white, yellow, black*. The number of cables in each image is randomly sampled from 1 to 10. A sample image from the dataset, along with its corresponding submasks and semantic mask, are portrayed in Figure 10.

## C ABLATION STUDY

Within the *prompt encoder network*, each block is fundamental to the proper sampling of point embedding vectors conditioned on text and performs a specific function in the process, as outlined in section 3.1.1. However, the *classifier network* architecture requires an in-depth ablation study. The modularity of our method permits the independent testing of the classifier and prompt encoder networks. We train multiple configurations of the classifier network and compare the classification accuracy that the network achieves, on the test dataset. In the remainder of this section, we discuss

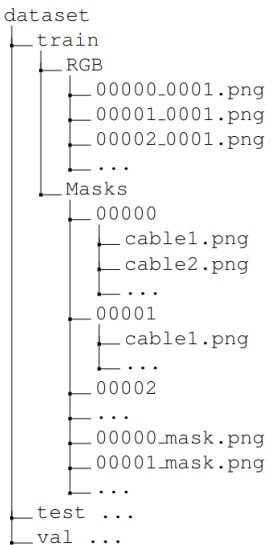

```
dataset
└─train
  ├─RGB
  │  ├─00000_0001.png
  │  ├─00001_0001.png
  │  ├─00002_0001.png
  │  └─...
  └─Masks
     ├─00000
     │  ├─cable1.png
     │  ├─cable2.png
     │  └─...
     ├─00001
     │  ├─cable1.png
     │  └─...
     ├─00002
     ├─...
     ├─00000_mask.png
     ├─00001_mask.png
     └─...
├─test ...
└─val ...
```

Figure 9: Dataset directory tree

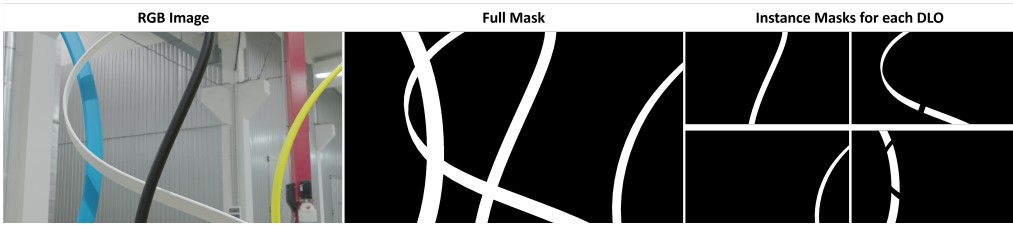

Figure 10: Dataset Example

various configurations of the classifier network and the labelling accuracy they achieve. The results are summarized in the Table 5.

The mask token that SAM outputs, contains encoded information about each of the $N$ submasks. Using this information, we need to generate tokens that can be used to classify each submask as *keep* or *discard*. Each of these $N$ tokens must be correlated with each other, which will extract information about duplicate masks. For this purpose we must use a self attention layer. The first study (**config 1** in Table 5) was conducted by passing these tokens through a self attention layer followed by a classifier MLP. This architecture caused the classifier network to diverge.

In the second study (**config 2** in Table 5), we introduced $N$ trainable tokens (we call these tokens *classifier tokens* in the remainder of this section). These tokens are first passed through a self-attention block, followed by cross attention with the mask tokens generated by SAM and finally through a classifier MLP. The cross attention block was added to fuse the submask information with the classifier token information. This method converged to a final binary classification accuracy of 76.37%.

The third study was conducted similar to *config 2* except that we switched the order of self and cross attention. Namely in **config 3**, we first fuse the mask token information and classifier token information using a cross attention block, which is then followed by a self attention block and then a classifier MLP. This configuration achieved an accuracy of 81.18%. This shows that cross attention-based fusing before self-attention improves classification performance. However, by using this method, the classifier network does not have direct access to the text information and cannot be controlled using text prompts for example.

In the fourth configuration (**config 4** in Table 5), we no longer instantiate trainable tokens. We take the output of the *sampler attention* block from the *prompt encoder network* (Fig. 3), which are the text-conditioned prompt embedding vectors. These embedding vectors are then fused using cross attention with mask tokens, followed by self attention and then a classifier MLP. This method achieves an accuracy of $80.26\%$, but improves *config 3*, by providing the classifier network with direct access to text information.

In the final configuration (**config 5** in Table 5), we use the classifier network exactly as portrayed in Fig. 3. Here we transform the text-conditioned prompts using an MLP, before passing it through the cross and self attention blocks. This method boosts the classifier performance significantly, with a classification accuracy of $84.83\%$. The comparison of all the aforementioned configurations is summarized in Table 5

| Configuration | Binary Classification Accuracy [%] |
|---|---|
| config 1 | NA (model diverged) |
| config 2 | 76.37 |
| config 3 | 81.18 |
| config 4 | 80.26 |
| **config 5** | **84.83** |

Table 5: Ablations of the classifier network (values in bold signify best performance)

## D  MORE QUALITATIVE RESULTS

Figures 11 and 12 show more instance segmentation masks generated by ISCUTE, on datasets from RT-DLO and mBEST as well as on our generated test dataset, respectively. All the images are examples of masks generated **without oracle**.

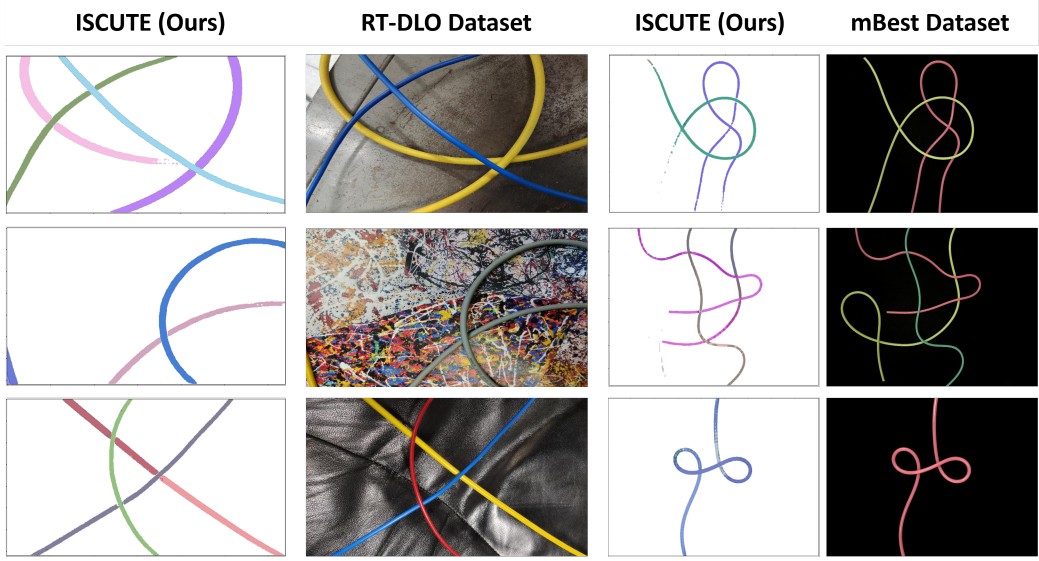

Figure 11: Qualitative results on RT-DLO and mBEST images

## E  LIMITATIONS OF FOUNDATION MODELS

The Oracle test results from Tables 1 and 2 display the maximum one-shot performance that can be achieved with the ViT-L/16-based version of SAM on the DLO-dataset, with up to 3 points in a prompt. The mask refinement method presented in SAM (Kirillov et al., 2023) can potentially improve this upper bound, but for a practical use-case, one-shot mask generation is desirable. CLIPSeg

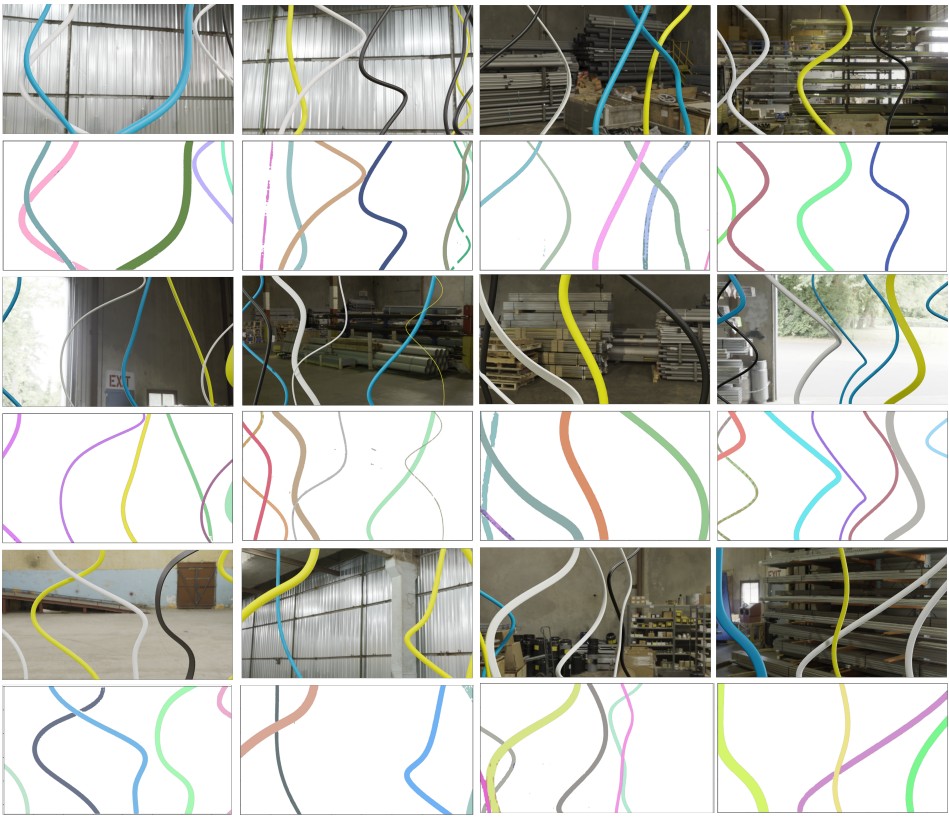

Figure 12: Qualitative results on our test images

also has some limitations on generating the perfect heatmap embedding for a given text prompt. This is demonstrated in Figure 13, where for the same prompt "cables," the model successfully locates all the cables in the bottom image, while it fails to locate the white cable in the top one.

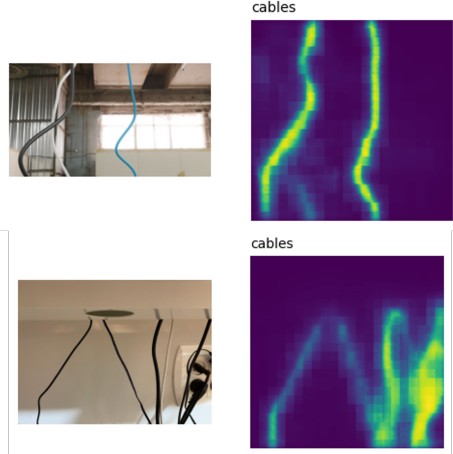

Figure 13: CLIPSeg generated heatmaps. In the top image, CLIPSeg is unsuccessful in detecting the white cable, while it detects all the cables in the bottom one. The prompt used was "cables".

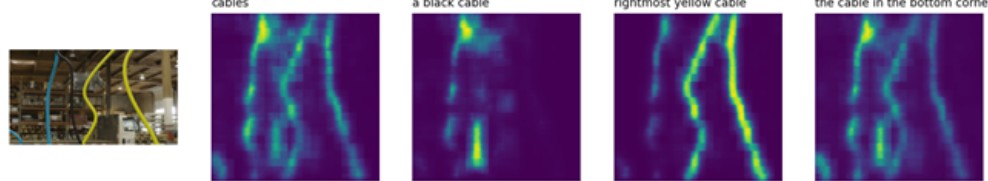

Figure 14: CLIPSeg's effectiveness on compliated text-prompts

