# OpenReview forum: "ISCUTE: Instance Segmentation of Cables Using Text Embedding"
_ICLR.cc/2024/Conference — Submitted to ICLR 2024_

### Official Review · Reviewer_NWxv · 2023-10-29

**Soundness:** 2 fair
**Presentation:** 2 fair
**Contribution:** 1 poor
**Rating:** 3
**Confidence:** 5

**Summary:**

This paper proposes an adapter model for encoding the text prompts to point prompts and filtering the masks generated by SAM. It achieves 91.21% mIoU on the DLO benchmark.

**Strengths:**

- Proposed a prompt encoder network to obtain point prompts from text prompts by CLIPSeg.
- Proposed a binary classifier network for the quality of SAM-generated masks.
- Achieved a solid result.

**Weaknesses:**

- Utilizing the combination of two powerful models, CLIPSeg and SAM, may be effective but not novel.
- The design motivation in the 3.1.2 section (i.e., MLP, cross-attention, self-attention) is missing.
- Few baselines. The only other method mentioned is RT-DLO. Considering the author is leveraging strong semantic segmentation methods, including SAM, they should compare their method with those segmentation methods.
- No ablation studies were conducted.

**Questions:**

- Please explain the motivation for components proposed in the 3.1.2 section.
- Please compare the proposed method with SAM, CLIPSeg, and other strong segmentation models by inference on DLO benchmarks.
- Please conduct ablation studies, including quantitative and qualitative analysis.

---

> ### Author Response · Authors · 2023-11-16
>
> We are sincerely grateful for your constructive comments and suggestions, as they play a crucial role in helping us improve our paper.
> We would like to draw attention to the fact that the novelty of our method lies in the adapter model we introduced. As discussed extensively in Sections 1 and 2.2, CLIPSeg and SAM are not capable of performing one-shot instance segmentation out-of-the-box, with the former one unable to perform instance segmentation at all and the latter requiring a laborious process of manual prompting. Furthermore, quoting from the SAM paper (section 8: limitations) “While SAM can perform many tasks, it is unclear how to design simple prompts that implement semantic and panoptic segmentation.” Another limitation of SAM is that it does not introduce any method to filter masks we do not need. The main novelty we introduce, overcomes both these limitations by presenting a simple way to prompt it, as well as a method to filter out masks that we do not require.
> Another point we wish to highlight is that we use the output of the PENULTIMATE layer of CLIPSeg and not the final semantic segmentation it generates. Our presented prompt encoder network transforms this embedding space directly into SAM’s prompt embedding space, without the need to sample individual pixels.
>
> Q1) We wish to thank you for pointing out a valid deficiency in our paper. We have carefully incorporated your suggestions in the revised paper. We have added a reference to Appendix C (Ablation study) in section 3. This appendix discusses the design motivation for each individual block, as well as the impact of removing it from the presented model. We wish to note that all the additions and changes were made based on extensive experiments carried out in each configuration.
>
> Q2) The RT-DLO method compares DLO-specific instance segmentation methods with other non-DLO specific baselines and shows a major improvement in comparison. We present a method that exceeds all the previously presented DLO-specific instance segmentation models, including RT-DLO, mBEST, FASTDLO, etc. (refer to table 2).
> Additionally, it's crucial to highlight that the CLIPSeg model does not execute instance segmentation, as detailed in Section 1. Similarly, SAM lacks a direct mechanism for DLO instance segmentation without manual point prompting. The generation of instance masks using this method heavily depends on the precision of point prompts, as depicted in Figure 2 of our paper. This direct application is also impractical in real-world scenarios for the same reasons.
> Another notable concern is the shortage of DLO instance segmentation baselines for meaningful comparisons. Due to the novelty of this field, there is a lack of robust datasets available for comprehensive research. This scarcity underscores the importance of the dataset we introduce alongside our method, as we anticipate it will expedite progress in DLO instance segmentation research.
>
> Q3) Thank you for your constructive comments and suggestions, and they are exceedingly helpful for us to improve our paper. We have carefully incorporated them in the revised paper. Please find the updates in Appendix C “Ablation Study”.

---

> ### Comment · Area_Chair_d6xS · 2023-11-21
>
> Reviewer NWxv,
>
> As the authors' rebuttal had been submitted, does it address your concerns?
>
> Please reply to the rebuttal, and pose the final decision as well.
>
> AC

---

> ### Comment · Reviewer_NWxv · 2023-11-22
>
> Thanks for AC’s kind reminder, and thanks for the authors’ response.
>
> I appreciate the authors’ effort in conducting ablation studies for their methods and making clarifications of their novelty. My concerns remain the following aspects.
> - The importance and contribution of DLO [3] and the proposed DLO-specific task. Though authors have emphasized the challenges of segmenting tiny objects like cables, they do not state the motivation or importance of this task. Moreover, they further propose a more specific DLO domain (i.e., cable). In other words, how will this task-specific instance segmentation, DLO, contribute to the community? What are the technical merits?
> - The incompleteness of the paper. The motivation of their proposed adapter (where “their novelty lies in” as they stated in the response) and ablation studies were missing in their main paper. Despite the authors adding the motivation of their proposed adapter and ablation studies during rebuttal, these major modifications may need another round of review. For example, they describe the motivation by analyzing the impact of adding or removing modules (ablation studies). This kind of motivation description from technical aspects needs to be considered carefully.
> - Insufficiency in quantitative comparisons. The paper claims their superior performance in mIoU metrics. However, in their main experiments (table 2), they only evaluate their method under DICE score to compare with other methods without explaining any reasons.
> - Unfairness in comparisons. As shown in Table 2, the proposed method achieves higher DICE score by a large margin compared with other DLO methods. However, for qualitative comparison (Figure 5), the authors present their segmentation results with white backgrounds, while demonstrating the groundtruth and all the other methods with black backgrounds. White backgrounds provide lower contrasts than black backgrounds. If we zoom in and observe the cases they provide carefully, we may notice that in the qualitative examples they provide, their proposed ISCUTE method is not always better than mBest [1] or RT-DLO [2]. In this case, considering the advantage in Table 2, the authors should explain the reason for using unfair background colors for qualitative comparisons.
> - Less importantly, I am also curious about the future work for this domain-specific task, as the proposed method achieves ~98% DICE score.
>
> I am willing to raise my score as the authors did respond and address some of my previous concerns, such as no ablation studies. However, I still lean toward a negative decision based on the major incompleteness, ambiguous contribution and importance, and relatively insufficient and potentially unfair comparisons.
>
> Thanks again for the authors’ effort and AC’s attention. I would like to humbly suggest the distinguished AC consider the contribution of the task, the completeness of the paper, and the experiments when making the final decision.
>
> [1] Alessio Caporali, Kevin Galassi, Bare Luka Zagar, Riccardo Zanella, Gianluca Palli, and Alois C. Knoll. RT-DLO: Real-Time Deformable Linear Objects Instance Segmentation. IEEE Transactions on Industrial Informatics, 2023. ISSN 19410050. doi: 10.1109/TII.2023.3245641.
>
> [2] Andrew Choi, Dezhong Tong, Brian Park, Demetri Terzopoulos, Jungseock Joo, and Moham- mad Khalid Jawed. mBEST: Realtime Deformable Linear Object Detection Through Minimal Bending Energy Skeleton Pixel Traversals. 2 2023. URL http://arxiv.org/abs/2302. 09444.
>
> [3] Riccardo Zanella, Alessio Caporali, Kalyan Tadaka, Daniele De Gregorio, and Gianluca Palli. Auto-generated Wires Dataset for Semantic Segmentation with Domain-Independence. In 2021 International Conference on Computer, Control and Robotics, ICCCR 2021, pp. 292–298. Institute of Electrical and Electronics Engineers Inc., 1 2021. ISBN 9781728190358. doi: 10.1109/ICCCR49711.2021.9349395.

---

> > ### Author Response · Authors · 2023-11-22
> >
> > We thank the AC for their kind reminder and appreciate the concerns laid out by the reviewer.
> > - The first block in the pipeline for automated robotic assembly and disassembly is detection of the target, followed by reaching->grasping->manipulating. To succeed in the aforementioned downstream tasks, detection needs to be robust, our method poses to address this objective in the assembly task. In addition, although the proposed method is task specific, with additional research offers a simple way to prompt and control the SAM to perform a myriad of segmentation tasks, which it previously lacked (as we discussed in our response).
> > - The motivation for each independent network has been discussed in the paper as well as our comments. The proposed prompt encoder network consists of just 4 single-layer blocks, each performing a subprocess in the main process of point prompt filtering. The proposed classifier network is initially motivated by similar classifier networks introduced in works such as DETR[1] and MaskFormer[2], where trainable tokens were used to classify N+1 object classes.
> > - The reasons for using DICE to compare to other methods has been mentioned in our initial submission - "Adhering
> > to recent testing protocols in the domain of DLO instance segmentation, as conducted by Choi et al.
> > (2023) in mBEST, we use the DICE score as a metric to analyze the performance of our model."
> > - The choice of a white background is not an intentional one, it is a byproduct of using the same mask plotting code provided by the Segment Anything Model. It is more of an aesthetic, default and standard choice in presenting instance masks. The shortcomings of models such as mBEST and RT-DLO are in assumptions about the specificity of scenarios. For example, occlusions, complex intersections involving multiple DLOs, generalization to various thicknesses and shape of the DLO cross-section and a strong reliance on obtaining an initial binary mask are some limitations faced by both models. Our model addresses and solves issues related to these limitations, as is showcased in the qualitative results on "in-the-wild" images as well as qualitative and quantitative results on datasets such as C1, C2 and C3. As our method currently stands, certain scenarios with complex self-loops are a limitation, especially without the use of the oracle method, as can be inferred from Table 2 (columns A and B, as oracle is not relevant during testing) and some of the qualitative results, as the respected reviewer correctly addresses. However, in the application of our method to industrial robotics, this limitation does not cause failure modes that often. Furthermore, these results are based on zero-shot transfer and we believe that few-shot fine-tuning with images of cables containing numerous loops, could alleviate these shortcomings.
> > - As discussed in bullet 1, this method has applications in an automated robotic assembly-disassembly pipeline. We are also exploring rigid object instance segmentation, where our method can be generalized to provide a simple and user-friendly way to prompt SAM.
> >
> > We wish to thank the AC once again for their involvement in promoting the discussion. We also wish to thank the reviewer for their accurate and valid concerns and hope that our response clears them out.
> >
> > Although all the concerns raised are valid, we would kindly like to highlight the fact that most, if not all of them can be resolved in a more satisfactory manner. Given that the deadline is today, if any concerns still remain after that, they can be fixed for the final manuscript. We hope that this fact is taken into consideration by the esteemed AC in their final decision.
> >
> > [1] Nicolas Carion, Francisco Massa, Gabriel Synnaeve, Nicolas Usunier, Alexander Kirillov, and Sergey Zagoruyko. End-to-End Object Detection with Transformers. 5 2020. URL http://arxiv.org/abs/2005.12872.
> >
> > [2] Cheng, Bowen, Alex Schwing, and Alexander Kirillov. "Per-pixel classification is not all you need for semantic segmentation." Advances in Neural Information Processing Systems 34 (2021): 17864-17875.

---

### Official Review · Reviewer_UDha · 2023-10-30

**Soundness:** 3 good
**Presentation:** 3 good
**Contribution:** 3 good
**Rating:** 6
**Confidence:** 3

**Summary:**

This paper proposes a novel structure for DLO instance segmentation, taking advantages of SAM and CLIPSeg. Via an adapter model, the proposed method can provide SAM with proper prompts for generating DLO masks. The overall framework achieves state-of-the-art performance on DLO-specific datasets, providing a new direction of solving DLO segmentation problems.

**Strengths:**

（1）The proposed method combines SAM with text conditions, and constructs a prompt encoder to help improve the overall DLO segmentation abilities.
（2）The proposed method achieves state-of-the-art performance compared with other recent algorithms on DLO instance segmentation.

**Weaknesses:**

（1）It seems that the proposed method relies on the assumption that if properly prompted, SAM can always provide correct cable segmentation masks. As the authors claimed, the performance upper-bound is limited by SAM and CLIPSeg. I wonder what is the exact upper-bound of these two methods, and how close can the proposed method reach this bound?
（2）Run-time for each method is not evaluated and analyzed in Table 1 and 2.
（3）Typo in Section 2.2: ” ... that have historically have been difficult to segment ...”

**Questions:**

(1) What is the exact upper-bound of these two methods, and how close can the proposed method reach this bound?
(2) How much time  does the proposed method need to take for DLO segmentation?

---

> ### Author Response · Authors · 2023-11-16
>
> We appreciate your valuable feedback and suggestions; they greatly contribute to enhancing our paper. The typos mentioned in the comments have been fixed in the updated PDF.
>
> Q1) This is a great question, and it needs some more exploration. From our experiments (results in table 1 “Oracle”), we observed that with the proposed architecture and using both the models in their frozen state, the method achieves an mIoU of 92.5% and 92.1% with and without augmentation respectively. We trained this method multiple times (by increasing the representation power and depth of the prompt encoder network as well) and tested it under the oracle setup. The variance of the results was very low. In table 2 as well, the “A+O” and “B+O” configurations have very close results across the board, leading us to conclude that this is an upper bound on the performance our method can achieve, which can only be traced back to the frozen foundation models.
>
> Q2) The runtime of our method is 330[ms], per image (1920x1080) on a single Nvidia RTX 2080 GPU; and 250[ms] on a single Nvidia A5000 GPU. This can mainly be attributed to the image encoder in SAM. We would like to add that unlike other baselines mentioned in our paper, our model’s runtime is independent of the number of DLOs or DLO intersections in the image. The method we propose focuses mainly on quality improvement of the instance segmentation of DLOs as well as the generalization capabilities of the model, using classical computer vision methods

---

> ### Comment · Area_Chair_d6xS · 2023-11-21
>
> Reviewer UDha
>
> As the authors' rebuttal had been submitted, does it address your concerns?
>
> Please reply to the rebuttal, and pose the final decision as well.
>
> AC

---

### Official Review · Reviewer_Bk1o · 2023-11-01

**Soundness:** 3 good
**Presentation:** 3 good
**Contribution:** 2 fair
**Rating:** 6
**Confidence:** 4

**Summary:**

This paper proposes an instance segmentation framework of cables built on top of CLIPSeg and segment anything model (SAM). By adding learnable adapters for prompt and class, the proposed model is able to achieve good zero-shot generalization capability. Experiments on several datasets show that the proposed approach outperforms several existing methods by a large margin and is relatively robust under different parameter settings.

**Strengths:**

The proposed approach is relatively straightforward as it is a direct application of CLIPSeg and SAM models to a high specific domain. The main idea of adding adapters is technically sound and also aligns well with the problem setting as in nature well labeled cable images are not easy to acquire. This validates the choice of using adapters in this approach instead of a full fine-tuning. In this sense, the proposed approach is reasonably motivated.

In addition, adding text prompt to the model allows for more flexibility compared to a vision only model. This also improves the zero-shot generalization.

Despite its simpleness, the proposed approach already works well on some data and outperforms existing approaches.

**Weaknesses:**

This work is overall good without significant flaws, but I do want to mention that it is more an application of existing models to a new domain with some modifications than a novel approach. The way of using these models is relatively straightforward. However, there still are a few questions to be answered. Please see detailed comments below.

**Questions:**

- Although the experiments have shown that the proposed approaches work well in many cases, it is still unclear how well it can generalize. The dataset used for evaluation consists of only 4 colors and all images are high resolution (1920x1080), and cables are placed at a similar distance to the camera and there is limited appearance variation. This can be seen from both training and validation data. This simplifies the problem a lot and can affect the performance of the model when it is tested on more realistic scenarios. For example, when there are "cables in the wild" which have diverse colors and are twisted with each other, or far away from the camera, or have large appearance variance in the training and testing sets, the proposed model may fail. It would be helpful to see how the model performs under this case, so that readers have a better understanding of its behavior on different data.

- The text prompts evaluated are quite limited. Only 3 choices are used, which seem not have enough coverage. In addition, all the 3 text prompts are single words that behave as class labels. Given that both the CLIPSeg and SAM model are very strong at recognizing a broad range of textual concepts. I would like to know how the proposed model reacts towards more complex, detailed text prompts - whether this could improve or reduce the model quality.
In page 7, the authors claim "we observe that the model generalizes better if it is trained using augmentations in the dataset". However, the improvement is very marginal on some data, e.g., from 97.01 to 97.71, which is larger on some other data. Any explanation to that?

- Some sentences are broken:
   - In Section 2.2, "However,Struggingle to" should be corrected.
   - In Section 2.2, "its prompt encoder with a single capable" should be corrected.

---

> ### Author Response · Authors · 2023-11-16
>
> Thank you for your constructive comments and suggestions, and they are exceedingly helpful for us to improve our paper.
> Q1) The generated data indeed shares a similar nature, featuring cables selected from a color range commonly found in industrial environments, including hues like yellow and black. Each image adheres to a standardized resolution of 1920x1080 and has served as the basis for our training. The results presented in Table 1 presents the outcomes derived from this dataset.
> Nevertheless, we wish to highlight the findings outlined in Table 2. The datasets referenced as C1, C2, and so forth, originate from the RT-DLO and mBEST papers. Notably, our model has not been exposed to or trained on these datasets that exhibit scenarios, colors, and shapes of DLOs unlike our training dataset. The "C[i]" datasets maintain a resolution of 640x360, while the "S[i]" datasets contain images of size 896 × 672. The colors of DLOs in these images as well as the backgrounds and scenarios of these images also differ from those we generated, as depicted in figures 4 (a), (b), (c), 5, and 11.
> Additionally, in figure 6, we present results pertaining to "cables in the wild" scenarios, showcasing genuine cable images captured using a smartphone camera in our laboratory. Employing our trained model, we conducted a "zero-shot" transfer to all the aforementioned scenarios.
>
> Q2) As mentioned in section 6, we are currently in the process of experimenting the effectiveness of our method to standard rigid-object instance segmentation. This involves training a general instance segmentation model with a wide variety of text-image pairs, as done in CLIP. Another direction we are currently exploring is the application of our method to more specific prompts such as “blue HDMI cable”, “black LAN cable”, etc. The presented research is aimed to solve one of the major problems in robotics research (DLO instance segmentation). We harness the power of SAM and CLIPSeg, with novel adapter model to control SAM’s segmentation output and to prompt it effectively.
> Furthermore, although CLIPSeg is capable of processing more complicated text prompts, it is most effective on simple “class-like” text prompts as you can see in the Figure 14 of our updated PDF.
>
> The influence of augmentation on generalizability is closely tied to the types of augmentations employed in our approach. During training, we adopt a randomized selection of augmentations in the pixel domain, encompassing techniques like grayscale, Gaussian blur, color jitter, and more. These augmentations play a crucial role in improving the model's ability to generalize across images with varying resolutions and diverse colors of DLOs, up to the upper limit set by SAM's performance. Notably, results without augmentation already closely approach this upper bound, indicating limited additional improvement with augmentation.
> However, the challenge lies in generalizing to the intricate shapes of DLOs, particularly those with numerous loops, which cannot be addressed through image-based augmentations. This presents an opportunity to highlight the complexities in segmenting DLOs compared to more rigid objects.
> Additionally, these findings can indicate that the dataset we generated may be reasonably homogeneous and reasonably representative of the data distribution in question. The details of this dataset, as outlined in the paper, will be made available upon publication.
>
> Q3) Thank you for pointing it out. These typos have been fixed.

---

> ### Comment · Area_Chair_d6xS · 2023-11-21
>
> Reviewer Bk1o,
>
> As the authors' rebuttal had been submitted, does it address your concerns?
>
> Please reply to the rebuttal, and pose the final decision as well.
>
> AC

---

### Author Response · Authors · 2023-11-21
**Rebuttal discussion reminder**

We wish to thank all the reviewers for their time and insightful comments about our paper. Following your suggestions we have made some improvements to our submission and we wanted to get more insights and feedback from you, so that we can submit the best version of our work.
We are open to further discussions and feedback about our rebuttal. We would like to draw the reviewers’ attention towards the same. But given that the deadline is tomorrow, we would kindly appreciate any discussion and feedback as soon as possible.
Thank you all in advance.

---

### Comment · Area_Chair_d6xS · 2023-11-21

Reviewers,

This is a reminder to reply to the authors ASAP.

The rebuttal had been posted for a while and we are close to the discussion due date.

AC

---

### Meta-Review · Area_Chair_d6xS · 2023-12-09

**Metareview:**

The paper presents an approach for instance segmentation of Deformable Linear Objects (DLOs) in robotics and automation, combining CLIPSeg's text-conditioned segmentation and SAM's zero-shot generalization for user-friendly, high-performance DLO identification, complemented by a dedicated, diverse dataset for this purpose. There are two reviewers giving positive scores and one votes for rejection. The general concerns from the reviewers are the technical contributions of the work by applying SAM-like approach on a particular domain, and the experiments are not very convincing yet in their comparisons. After carefully reading the paper, the rebuttal, and the discussion. The AC finds this paper might be a better fit for vision application conferences/journals, but not be a good fit for the ICLR community. The AC recommends rejection for this paper.

**Justification For Why Not Higher Score:**

From the discussions, the reviewers were still concerning about the novelty, and the insufficiency in quantitative comparisons. The AC agreed on these aspects.

The AC found this paper might be a better fit for vision application conferences/journals. For ICLR community, e.g., as a submission focusing on representation learning, further improvement is still needed.

**Justification For Why Not Lower Score:**

N/A.

---

### Decision · Program_Chairs · 2024-01-16

Reject